# Environment-Adaptive Object Detection Framework for Autonomous Mobile Robots

**DOI:** 10.3390/s22197647

**Published:** 2022-10-09

**Authors:** Donghun Shin, Joongho Cho, Jaeho Kim

**Affiliations:** Department of Electrical Engineering, Sejong University, Seoul 05006, Korea

**Keywords:** autonomous mobile robots, object detection, environment-context awareness, model-caching, lightweight scene classification, class imbalance

## Abstract

Object detection is an essential function for mobile robots, allowing them to carry out missions efficiently. In recent years, various deep learning models based on convolutional neural networks have achieved good performance in object detection. However, in cases in which robots have to carry out missions in a particular environment, utilizing a model that has been trained without considering the environment in which robots must conduct their tasks degrades their object detection performance, leading to failed missions. This poor model accuracy occurs because of the class imbalance problem, in which the occurrence frequencies of the object classes in the training dataset are significantly different. In this study, we propose a systematic solution that can solve the class imbalance problem by training multiple object detection models and using these models effectively for robots that move through various environments to carry out missions. Moreover, we show through experiments that the proposed multi-model-based object detection framework with environment-context awareness can effectively overcome the class imbalance problem. As a result of the experiment, CPU usage decreased by 45.49% and latency decreased by more than 60%, while object detection accuracy increased by 6.6% on average.

## 1. Introduction

In recent years, with the advancement of autonomous driving, advanced battery, and artificial intelligence technologies, robots have been developed into autonomous mobile robots (AMRs) that can autonomously perform assigned missions while moving through various spaces. Autonomous driving, obstacle avoidance, and object detection are essential functions that enable AMRs to perform their tasks efficiently. In particular, object detection technology is one of the key technologies that forms the foundation for other parts. In recent years, various deep learning networks based on convolutional neural networks (CNNs) have been proposed for image-based object detection [1,2,3,4]. However, the proposed CNN-based object detection models are trained with a large-scale image dataset that is collected without considering the environment. When these models are applied to AMRs without considering the mission environment, object detection accuracy for robots carrying out missions in a specific environment decreases, resulting in failed missions, as shown in Figure 1. Such object detection errors may cause a security surveillance robot at an airport or a patient care robot in a hospital to fail in its mission.

Yin Cui et al. mentioned the class imbalance problem as a major cause of such object detection errors [5]. The class imbalance problem occurs when the occurrence frequencies of the classes in the training dataset are significantly different. Real-world datasets exhibit the distribution of a long-tailed dataset, as shown in Figure 2a, where a small number of classes constitute most of the data, and the remaining classes make up the minority. Hence, the class imbalance problem exists in real-world datasets, and this causes critical problems in the performance of the object detection models. For example, as shown in Figure 1a, the advertising panel advertising a car at an airport terminal is recognized as a car, which occurs with high frequency in the training data. In Figure 1b, the special light hanging on the wall in a hospital room is incorrectly recognized as a wall-mounted air conditioner. These problems occur with high frequency in the training data. In particular, this type of problem may become more serious for AMRs that are required to perform a mission targeting a special object in a specific environment.

Studies have been conducted to solve this class imbalance problem, and we mainly use two strategies: the re-sampling technique and the re-weighting technique. The re-sampling technique is a method that adds (oversampling) or removes (under-sampling) data according to the occurrence frequencies of classes [6,7,8]. In the oversampling method, overfitting can occur because of overlapping training data, which is a possible cause of the poor generalization performance of the trained model. Similarly, in the undersampling method, the accuracy of the trained model can decrease owing to insufficient training data. Because of these performance degradation problems, recent studies have been conducted on the re-weighting approach using a new loss function that works well with the long-tailed dataset with class imbalance. The re-weighting technique is a method that adjusts class imbalance in the training process by adjusting the loss function to assign high weights to the samples of the low-frequency classes and low weights to the samples of the classes that frequently occur [9,10]. Representative examples include the focal loss used in RetinaNet [11], class-balanced loss proposed by Y. Cui [5], and LDAM loss proposed by K. Cao [9].

Such a class imbalance shows different characteristics depending on the environment. As shown in Figure 2, the datasets for different specific environments will have different distributions of class imbalance. In addition, the order of occurrence frequencies of classes changes depending on the environment. For example, the class index “Pillow” occurs more frequently than “Luggage” in the overall environment, as shown in Figure 2a. However, in the airport terminal environment, the class index “Luggage” occurs much more frequently than “Pillow”, and the class index “Toilet” occurs very rarely, as shown in Figure 2b. Moreover, the distribution of class index “Pillow” is the highest in the hospital room environment, as shown in Figure 2c. Different distributions across these environments indicate that solving the class imbalance problem in the entire environment does not mean solving the problem in a specific environment. In other words, a model trained in the overall environment through the loss function with the re-weighting technique may have low object detection accuracy in a specific environment. Hereinafter, the problem of varying frequencies of occurrence of classes in the dataset depending on the environment will be referred to as *environment class imbalance.*

To overcome the environment class imbalance for an AMR performing its mission while moving in various environments, we propose the multi-model-based object detection framework with environment-context awareness (M-ODF). The M-ODF provides a systematic solution that effectively resolves the environment class imbalance problem by using multiple object detection models for each environment. To this end, we first present a training process that creates multiple object detection models based on each environmental context. Next, we use a lightweight screen classification method for environmental context recognition so that the AMR can select an object detection model according to each environmental context. Finally, we propose a model caching algorithm for efficient use of multiple object detection models. Through experimental results, we demonstrate that the M-ODF can effectively overcome the environment class imbalance problem.

The rest of this paper is organized as follows. Section 2 introduces previous research on CNN-based object detection and studies conducted to solve the class imbalance problem. Section 3 describes the mechanisms for creating multiple models and effectively using the models generated for each environment to mitigate the environment class imbalance problem. Section 4 analyzes the proposed method and the existing state-of-the-art deep learning model through experiments. Finally, Section 5 concludes the paper and discusses directions for future research topics.

## 2. Related Work

In recent years, various deep learning models based on CNNs have been proposed for object detection. The CNN-based object detection method is divided into a two-stage model that sequentially processes the region proposal (also known as the bounding box) and region classification, and a one-stage model that processes the region proposal and region classification in parallel [12].

The two-stage model first extracts candidate regions in which objects are likely to exist using region proposal. It then performs object classification using a CNN-based classifier to recognize objects in each region. The two-stage model takes more time because it needs to process the region proposal and region classification sequentially. R-CNN [13], Spp-Net [14], Fast R-CNN [1], and Faster R-CNN [2] are representative object detection models based on the two-stage model. The one-stage method that processes the region proposal and object recognition simultaneously was proposed to solve the limitations of two-stage-based object detection models, which are difficult to apply to real-time applications. YOLO [15], SSD [16], YOLOv2 [3], DSSD [17], and YOLOv3 [4] are representative one-stage object detection networks. Although the one-stage model improved the object detection speed, it has the problem of low accuracy. Recently, the latest versions of YOLO networks, such as YOLOv4 [18] and YOLOv5 [19], have greatly improved the accuracy. However, this still does not mitigate the problem of environment class imbalance, as shown in Figure 1. In this study, we use YOLOv5 as the base object detection network for our proposed M-ODF. Because the class imbalance problem is the main reason for the degradation of object recognition accuracy, studies have been conducted using re-sampling and re-weighting approaches.

In the initially proposed re-sampling approach, data for training are oversampled or undersampled according to the occurrence frequency of each class in the raw dataset [6,7,8]. More recently, re-weighting methods have been proposed that adjust the loss based on the class distribution of the dataset. Li et al. proposed a gradient harmonized single-stage detector to solve this imbalance problem [10]. They reported that the class imbalance problem can be summarized as the imbalance of the gradient norm. Their detector determines the weight of the samples according to the gradient density. Notably, Cao. et al. proposed a label-distribution-aware margin (LDAM) loss function [9]. LDAM resolved the class imbalance problem using the margin boundary of each class. Cui et al. also proposed a class-balanced loss based on the effective number of samples, defined as the volume of samples [5]. They designed the class-balanced loss function to address the problem of training on imbalanced data by introducing a weighting factor that is inversely proportional to the effective number of samples. However, they still failed to mitigate the problem of environment class imbalance caused by the difference in class distributions according to the environment.

## 3. Multi-Model-Based Object Detection Framework with Environment-Context-Awareness

To overcome the environment class imbalance problem, we propose the multi-model-based object detection framework with environment-context awareness (M-ODF) that efficiently uses multiple object detection models trained in each environmental context. Figure 3 shows the object recognition process of M-ODF while an AMR moving through various mission spaces (e.g., spaces in a shopping mall) performs tasks. Figure 3a (object detection step) shows the process of object detection of AMR. The object detection function is triggered in every input video frame to support real-time applications. In this process, object detection is performed using a model corresponding to the environmental context of the space in which the AMR is currently performing its mission. Figure 3b (environmental context recognition step) shows the process of recognizing the environmental context for the space in which the AMR is performing its mission. A scene classification technique is used to recognize the environmental context. We also propose a lightweight scene classification scheme to reduce the computing resources. Finally, Figure 3c (model caching step) shows the model-caching process to quickly switch the object detection model when the environmental context of AMR is changed. We propose a transition probability-based model caching mechanism in which AMR preferentially caches the model corresponding to the environmental context of the neighboring space with high movement probability.

In this section, we describe the main algorithms of M-ODF that are used to efficiently create and utilize multiple models; that is, multi-model-based object detection, lightweight scene classification, and transition-probability-based model caching.

### 3.1. Multi-Model Based Object Detection

Previous studies used re-sampling and re-weighting methods to overcome the class imbalance problem. These studies resolved the class imbalance problem using only a single object-based model. Therefore, we will refer to the previous methods collectively as a single model-based object detection framework (S-ODF). S-ODF cannot solve the problem of environment class imbalance, in which the frequency of occurrence of classes in a dataset changes according to the environment. To tackle this issue, we propose M-ODF, which mitigates the environment class imbalance by effectively utilizing the multiple object detection model trained according to the environment.

Environment-specific datasets are required to train object detection models specialized for each environment. Figure 4 shows the procedure for training environment-specific object detection models. To obtain object detection models for each environment, we classify the training dataset by environmental context and train each model using the divided datasets. First, as shown in Figure 4a, we apply a scene classifier to the training dataset, which is not divided by the environmental context, to create environment-specific datasets. The class (e.g., airport terminal or bookstore) used in the scene classifier should be carefully selected to sufficiently represent the environment in which the AMR is carrying out its mission. The next step is the training process, wherein we train object detection models with environment-specific datasets. However, the size of an environment-specific dataset created by classifying according to the environment from the original is insufficient to train the object detection model. To handle the problem of the small training dataset, we use transfer learning, a machine learning method that reuses a pretrained model as the starting point for a model. Additionally, we use K-fold cross-validation to achieve data augmentation effects (Figure 4b).

### 3.2. Lightweight Scene Classification

To select an appropriate model according to the environment, it is necessary to accurately recognize the environmental context of the space in which an AMR is currently performing its task. Scene classification helps to determine the current environmental context. However, frequently performing scene classification causes the AMR to consume many resources. Therefore, we propose a lightweight scene classification method to alleviate the resource consumption of the AMR. The proposed scene classifier performs scene classification to recognize the environmental context only when a scene change is detected through the lightweight scene change detector. As a result, the proposed scene classifier uses fewer resources than the periodic scene classifier. The proposed lightweight scene classifier consists of scene change detection and a scene classification process.

The scene change detection process is further divided into image preprocessing and scene change detector modules. The image preprocessing module decreases the noise in the raw image from the camera and reduces the size of the image to help with lightweight scene change detection. The image processing first applies the Gaussian blur filter to reduce noise because the noise in the images collected from the camera may affect the scene change detection. Next, max pooling is used on images to which the filter has been applied to reduce the load on the scene change detection. By using max pooling, the sizes of the images are reduced while maintaining their characteristics. The scene change detector detects changes in the color and brightness between the previous and the current scene to detect the scene change. We convert the preprocessed image into grayscale and HSV (hue, saturation, value) images to recognize changes in color and brightness features. The grayscale image is used to represent the brightness and the HSV image is used to express the hue and saturation of the RGB image. The two feature images are transformed into histogram vectors. The histogram vectors are then compared with those of the previous scene, and a scene change is detected based on the amount of change in the histogram.

Figure 5 illustrates the process of recognizing the environmental context through lightweight scene classification and object detection using models stored in the model store. The lightweight scene classification performs scene change detection according to the preset interval for $T_{SC}$, as shown in Figure 3b. The scene change is determined based on the change of color and brightness feature values between the previous and the current scene. If the amount of change measured is larger than the predefined reference value, it is determined that the scene has changed. When a scene change is detected, scene classification is performed to recognize the new environmental context of the current space of the AMR. If the environmental context of the scene changes, object detection is performed using the new model corresponding to the new environmental context [20].

### 3.3. Transition Probability-Based Model Caching

The appropriate model for the environment should be cached in the GPU in advance to perform object detection in each environment. If the object detection model is not cached, delays may occur due to the time required to load the model. To address these problems, we propose transition-probability-based model caching. The basic idea of our proposed model caching is to precache the model for the environment corresponding to the neighboring space with the highest transition probability, which is the probability of moving from the current state to the next state. The transition probability is an element of the converged state transition matrix calculated based on the continuous movement of AMR in the mission space. The transition probability $P_{ij}$ used in the state transition matrix is the probability of moving from $E_{i}$ to $E_{j}$, as shown in Equation (Equation 1), and if it is not possible to move from $E_{i}$ to $E_{j}$, the transition probability is zero.
(1)$$P_{ij}=P\left[E_{i}|E_{j}\right]$$

When the AMR moves from $E_{i}$ to $E_{j}$, $P_{ij}$ is updated through Equation (Equation 2).
(2)$$P_{ij}=C_{ij}/C_{i}$$

Here, $C_{i}$ denotes the total number of environment change detections performed in state $E_{i}$, and $C_{ij}$ represents the number of transitions from state $E_{i}$ to state $E_{j}$.

From the converged state transition matrix, we can determine the highest transition probability of the AMR moving from the current state to the next state. Therefore, precaching the model in order of the highest transition probability increases the probability of using the cached model. The hit rate also increases as the cache size increases, and if the cache size is larger than the maximum degree (the number of edges in a state) of the state transition diagram, the hit rate will be 100%. The result of the state transition matrix learned by an AMR performing tasks in the mission environment (Figure 6a) is shown in Figure 6b, and Figure 6c illustrates the state transition diagram of the AMR. Figure 7 shows how the state transition matrix is calculated when the cache size is three in the environment shown in Figure 6a, and how the state transition matrix is utilized for model caching. As shown in Figure 7, in scenario (a), an AMR moves from $E_{d}$ to $E_{a}$ and then to $E_{b}$ ($E_{d}$ → $E_{a}$ → $E_{b}$) according to the path. The AMR first performs scene classification to recognize the environmental context of the current space ($E_{d}$). Thereafter, it loads the object detection model ($M_{d}$) suitable for the environmental context ($E_{d}$) into system memory and updates the transition probability $P_{dd}$ to one. In scenario (a), environment change detection occurs five times. Four out of five times, an environment change is not detected, and the state changes from $E_{d}$ to $E_{a}$ at the fifth environment change detection. Hence, $P_{dd}$ is updated to 0.8 and $P_{da}$ to 0.2. Moreover, in state $E_{a}$, the environment change detection was performed ten times. Nine out of ten times, the environment does not change, but the state changes from $E_{a}$ to $E_{b}$ for the tenth environment change detection. Hence, $P_{aa}$ is updated to 0.9 and $P_{ab}$ to 0.1. In this manner, the state transition matrix is continuously updated at each $T_{SC}$ interval. When the number of models cached in the AMR reaches the cache size, they are deleted from the cache in ascending order of transition probability.

## 4. Results and Discussion

### 4.1. Experiment Setup

We conducted experiments on scenarios of a guide robot assisting a visually impaired person in a shopping mall to analyze the performance of the proposed M-ODF. The shopping mall selected for the experiment is 80 m wide and 40 m long, and its composition is shown in Figure 8. The shopping mall consists of 19 spaces; that is, department_store, hospital_room, hotel_room, lobby, entrance_hall, bookstore, clothing_store, jewelry_shop, restaurant, shopping_mall_indoor, drugstore, pharmacy, pub_indoor, bar, candy_store, coffee_shop, florist_shop_indoor, gift_shop, and shoe_shop. In these experimental environments, the AMR moves at an average speed of 2 m/s and carries out missions.

### 4.2. Datasets and Models

As shown in Figure 4, we need a general environment dataset, scene classifier, and pretrained object detection model to create multiple object detection models according to the environment. We used the Objects365 [21] dataset as the general environment dataset, and the scene classifier was trained via transfer learning using ResNet [22] trained with Places365 [23]. The classifier classified the Objects365 data into environment-specific datasets. The pretrained model for environment-specific object detection models was created by training YOLOv5m [19] on the Objects365 dataset. The environment-specific object detection models were retrained for each environment-specific dataset using the pretrained model YOLOv5m. We used k-fold verification (K = 5) in the training process to solve the problem of insufficient data.

### 4.3. Experimental Results

#### 4.3.1. Accuracy of Multi-Model Based Object Detection

Figure 9 shows the results of the object detection accuracy using M-ODF and S-ODF for 19 spaces in experimental scenarios. Object detection accuracy was evaluated using mAP for intersection over union (IOU) of 0.5 and IOU of 0.5:0.95 (0.5 to 0.95, step 0.05). M-ODF improved the accuracy by at least 0.6%, at most 28%, and an average of 6.6% at IOU 0.5 compared to S-ODF. At IOU 0.95, M-ODF improved the accuracy by at least 1%, at most 21%, and an average of 6.2%. Compared to S-ODF, M-ODF displayed improved accuracy in all environments. Figure 10 shows an example in which M-ODF solved the environment class imbalance problem at the bookstore, drugstore, entrance hall, and lobby, and correctly carried out object detection in each environment. In the bookstore, M-ODF detected books and storage boxes, which S-ODF did not detect. In addition, M-ODF detected bottles and cups more accurately in the drugstore and detected chairs and lights more accurately in the lobby. In the entrance hall, the S-ODF incorrectly detected streetlights and mirrors, but the M-ODF did not.

#### 4.3.2. Performance of Light Weight Scene Classification

The data and parameters used in the experiment to evaluate the lightweight scene classification are as follows. We used ten videos lasting five minutes each for the experiment and defined the interval at which the image was collected, $T_{SC}$, as 1s, and the input image size as 125 × 125 pixels. In addition, the noise reduction filter used was cv2. GaussianBlur of 3 × 3 kernels provided by OpenCV. The scene classifier used in lightweight scene classification was created using a transfer-learning MobileNetV3_Large Network [24] using ImageNet [25] learning data. At this time, the last layer of MobileNetV3_Large Network was frozen and trained using images prepared for the experiment. Lightweight scene classification inference was carried out, and five scenes that were inferred to have the highest reliability among the scene classification results were selected, similar to conventional scene classification. Top1 is the case in which the most reliable scene matched the labeled data, and Top5 is defined as the case in which the labeled data matched any of the five scenes. The evaluated results of the accuracy of scene classification for 10 test videos are shown in Figure 11. Compared to conventional scene classification, both Top1 and Top5 showed error rates within 0.8%, and CPU usage decreased by 45.49%. The proposed model can achieve low CPU usage with similar accuracy because the lightweight scene classification module accurately detects the moment of scene change and performs the scene classification only when it detects a scene change.

#### 4.3.3. Efficiency of Transition Probability-Based Model Caching

We analyzed whether M-ODF efficiently utilizes multiple models through model caching in the experimental environment shown in Figure 8. First, the AMR randomly selected the start and destination points and moved 10,000 times to learn the transition probability matrix used in model caching. The resulting converged transition probability matrix is listed in Table 1. Table 2 lists the results of 100 tests, measuring the hit rate and latency using model-caching based on the transition probability matrix learned with the AMR moving between the randomly selected start and destination points. When using transition-probability-based model caching, even with the smallest cache size (cache size = 2), M-ODF increased the average hit rate by 7% and reduced latency by 0.2 s on average. This improvement was because the models with the highest probability of the next movement had been cached previously. In addition, as the size of the cache increased, the hit rate increased, and the latency decreased. In particular, hit rate and latency changed rapidly when the cache size changed from 1 to 2 and from 2 to 3. This result occurs because most of the edges of the state transition diagram have three or fewer degrees. When the cache size reaches 10, it caches all models for neighboring environments that can move from the current environment, resulting in a 100% hit rate.

## 5. Conclusions

We presented M-ODF as a systematic solution to overcome the problem of environment class imbalance, which greatly decreases the accuracy of object recognition of robots that perform tasks while moving in various environments. The M-ODF effectively resolves the environment class imbalance problem by using multiple object detection models suitable for each environment. To achieve this, we proposed three key techniques: muti-model-based object detection, lightweight scene classification, and transition probability-based model caching. Through experiments, we verified that our proposed M-ODF improved accuracy by 6.6% on average compared to the previous single-model-based object detection (S-ODF). As the industry of AMRs is growing significantly, the environment-adaptive object detection framework will represent a contribution, enabling AMRs to carry out their tasks accurately.

## Figures and Tables

**Figure 1 sensors-22-07647-f001:**
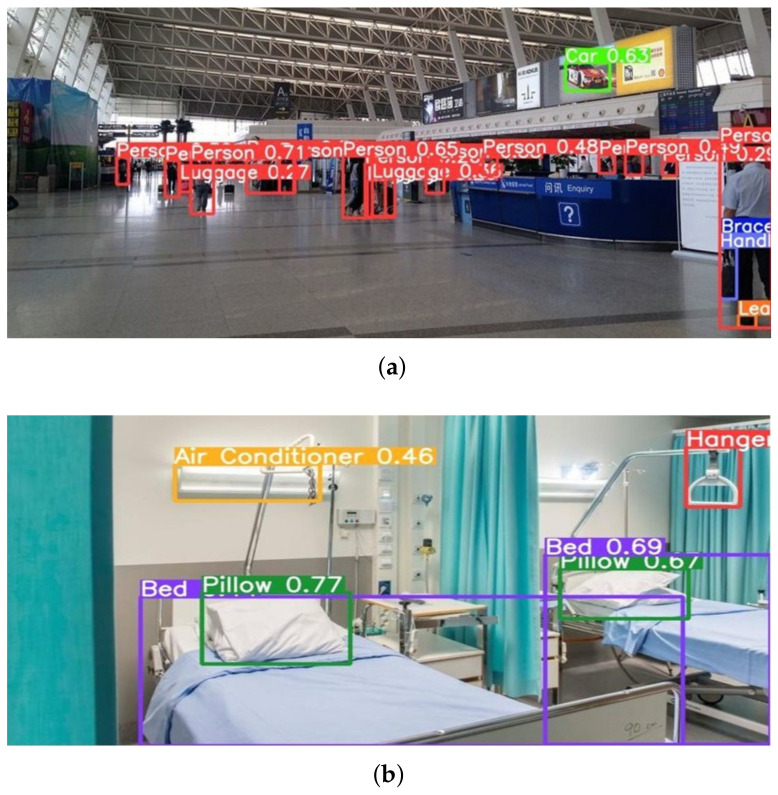
Incorrect object detection is performed in each environment. (**a**) A case in which the advertising panel at an airport terminal is highlighted in green and recognized as a car. (**b**) A case in which the light in a hospital room is highlighted in yellow and incorrectly recognized as an air conditioner.

**Figure 2 sensors-22-07647-f002:**
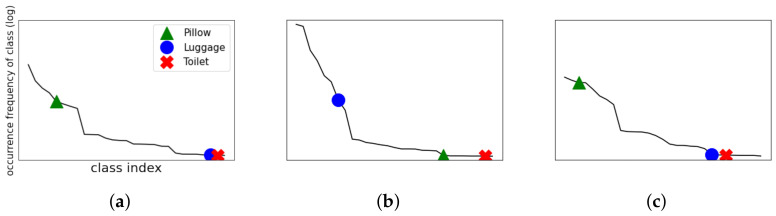
Class imbalance in different environments. Class distributions of (**a**) the entire Objects365 data, (**b**) airport terminal environment data, and (**c**) hospital room environment data.

**Figure 3 sensors-22-07647-f003:**
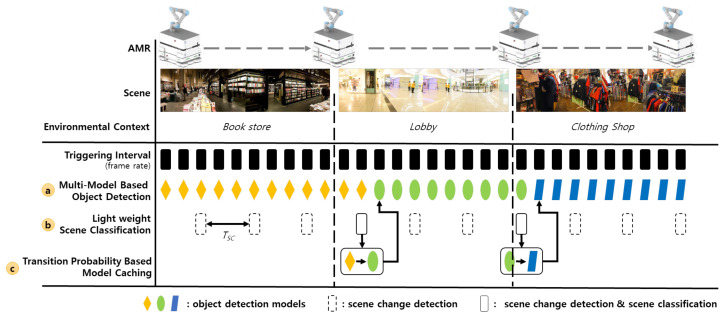
Overall process of M-ODF.

**Figure 4 sensors-22-07647-f004:**
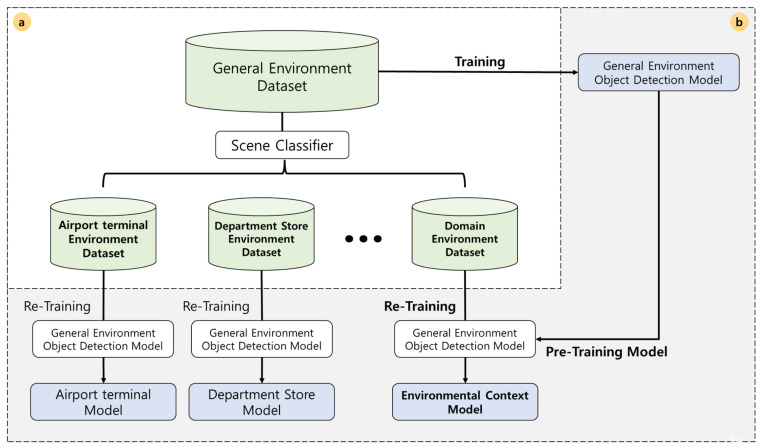
Procedure for developing an environment-specific object detection model.

**Figure 5 sensors-22-07647-f005:**
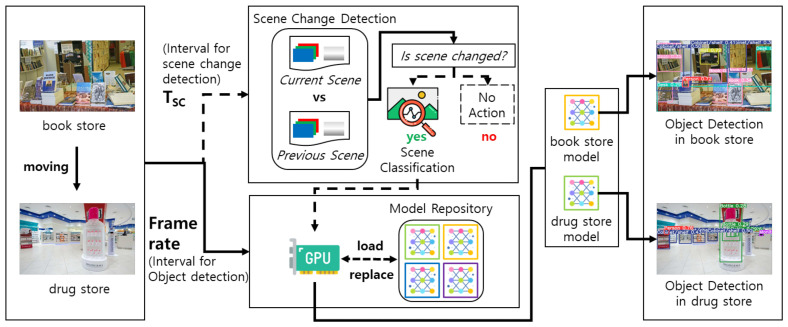
Process of lightweight scene classification in AMR.

**Figure 6 sensors-22-07647-f006:**
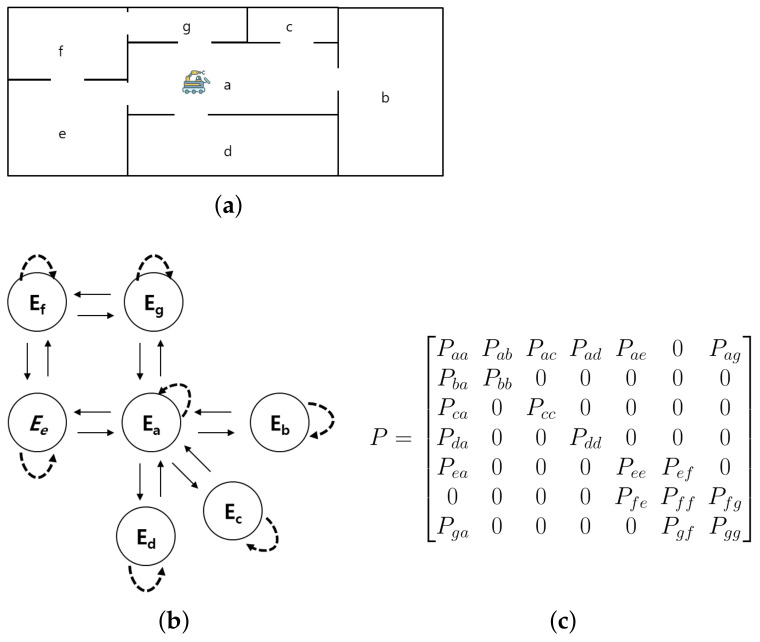
(**a**) Mission environment, (**b**) state transition diagram, and (**c**) state transition matrix for AMR.

**Figure 7 sensors-22-07647-f007:**
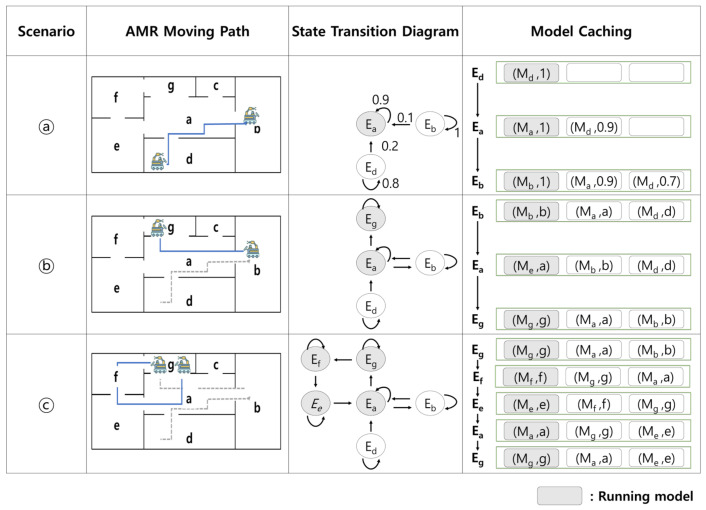
Example of transition probability-based model caching (cache size = 3).

**Figure 8 sensors-22-07647-f008:**
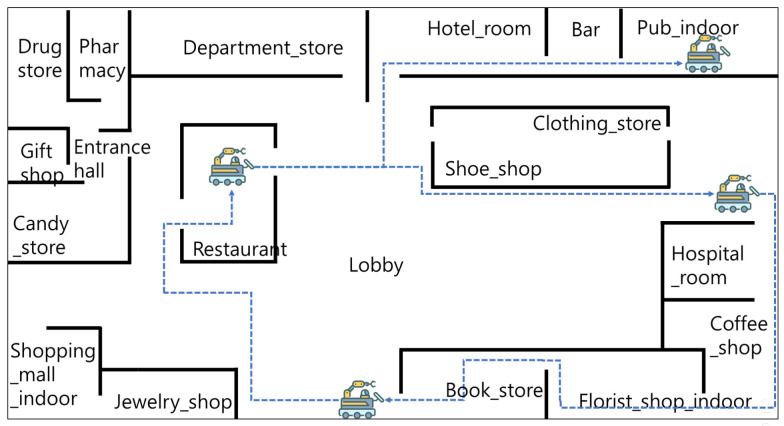
AMR experiment scenario.

**Figure 9 sensors-22-07647-f009:**
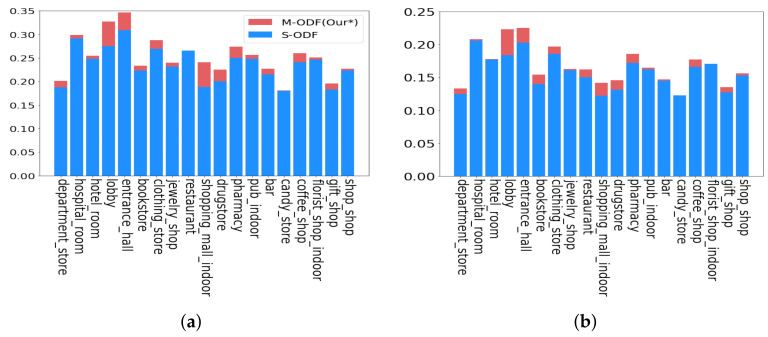
Accuracies of M-ODF and S-ODF. (**a**) mAPs for IOU of 0.5. (**b**) mAPs for IOUs 0.5–0.95 (step: 0.05).

**Figure 10 sensors-22-07647-f010:**
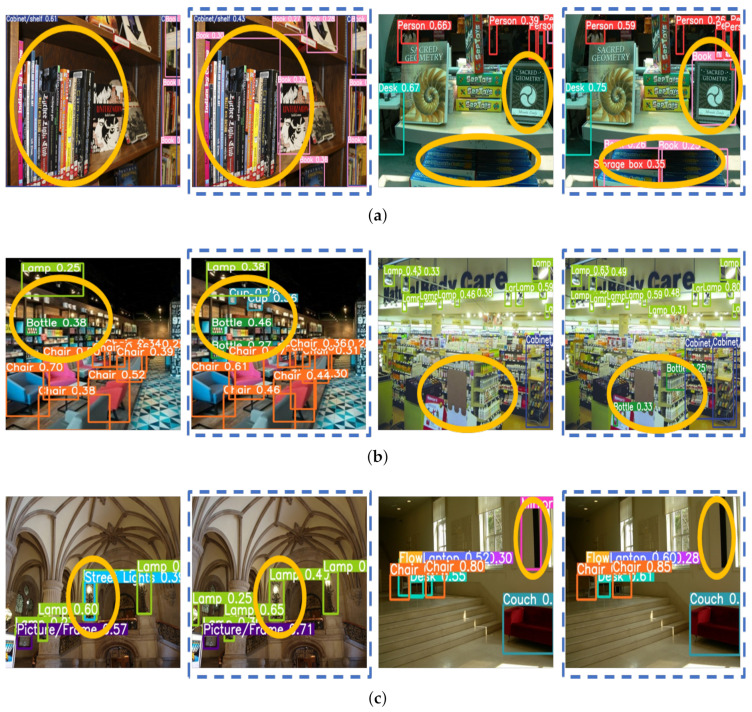
Object detection results of S-ODF (the image on the left from each pair of images) and M-ODF (image on the right with the dotted line from each pair of images) for each environment: (**a**) Bookstore, (**b**) Drugstore, (**c**) Entrance hall, and (**d**) Lobby.

**Figure 11 sensors-22-07647-f011:**
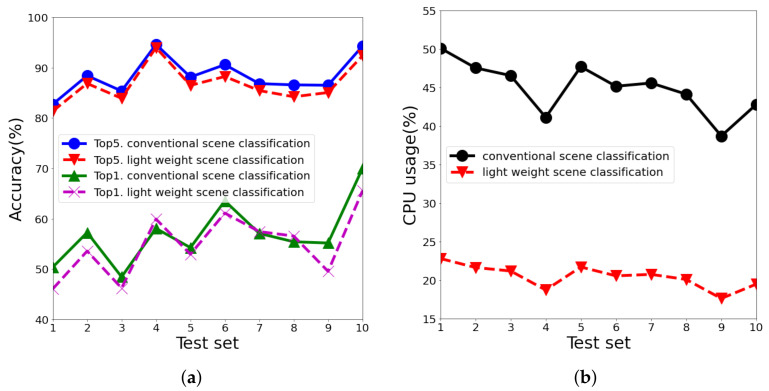
Performance analysis of lightweight scene classification: (**a**) Detection accuracies for Top1 and Top5, (**b**) CPU usages.

**Table 1 sensors-22-07647-t001:** State transition matrix of the shopping mall.

	Depart Ment _Store	Hospital _Room	Hotel _Room	Lobby	Entrance _Hall	Book Store	Clothing _Store	Jewelry _Shop	Restaurant	Shopping _Mall Indoor	Drug Store	Pharmacy	Pub Indoor	Bar	Candy Store	Coffee Shop	Florist Shop Indoor	Gift Shop	Shoe Shop
department_store	0.833	0	0	0.167	0	0	0	0	0	0	0	0	0	0	0	0	0	0	0
hospital_room	0	0.868	0	0.115	0	0	0	0	0	0	0	0	0	0	0	0.017	0	0	0
hotel_room	0	0	0.893	0	0	0	0	0	0	0	0	0	0	0.107	0	0	0	0	0
lobby	0.005	0.007	0.016	0.918	0	0.007	0.004	0.004	0.031	0.004	0	0	0	0	0	0	0	0	0.004
entrance_hall	0	0	0	0	0.837	0	0	0	0	0	0.04	0.041	0	0	0.042	0	0	0.04	0
bookstore	0	0	0	0	0	0.904	0	0	0	0	0	0	0	0	0	0	0.096	0	0
clothing_store	0	0	0	0.157	0	0	0.833	0	0	0	0	0	0	0	0	0	0	0	0.01
jewelry_shop	0	0	0	0.068	0	0	0	0.929	0	0.004	0	0	0	0	0	0	0	0	0
restaurant	0	0	0	0	0.094	0	0	0	0.906	0	0	0	0	0	0	0	0	0	0
shopping_mall_indoor	0	0	0	0.068	0	0	0	0.004	0	0.929	0	0	0	0	0	0	0	0	0
drugstore	0	0	0	0	0.022	0	0	0	0	0	0.978	0	0	0	0	0	0	0	0
pharmacy	0	0	0	0	0.025	0	0	0	0	0	0	0.975	0	0	0	0	0	0	0
pub_indoor	0	0	0	0	0	0	0	0	0	0	0	0	0.984	0.016	0	0	0	0	0
bar	0	0	0.011	0	0	0	0	0	0	0	0	0	0.07	0.918	0	0	0	0	0
candy_store	0	0	0	0	0.025	0	0	0	0	0	0	0	0	0	0.975	0	0	0	0
coffee_shop	0	0.115	0	0	0	0	0	0	0	0	0	0	0	0	0	0.878	0.007	0	0
florist_shop_indoor	0	0	0	0	0	0.006	0	0	0	0	0	0	0	0	0	0.104	0.89	0	0
gift_shop	0	0	0	0	0.032	0	0	0	0	0	0	0	0	0	0	0	0	0.968	0
shoe_shop	0	0	0	0.159	0	0	0.008	0	0	0	0	0	0	0	0	0	0	0	0.833

**Table 2 sensors-22-07647-t002:** Performance of transition-probability-based model caching.

Cache Size	Hit Rate	Latency (s)
1 (No Cache)	88.7%	0.33
2	95.7%	0.13
3	97%	0.09
4	97.4%	0.08
5	97.5%	0.07
6	97.5%	0.07
10	100%	0.00

## Data Availability

All datasets used for supporting the conclusions of this article are available from the public data repository as follow. Objects365: https://www.objects365.org/download.html (accessed on 25 August 2022). ImageNet: https://www.image-net.org/download (accessed on 25 August 2022). Places365: http://places2.csail.mit.edu/download.html (accessed on 25 August 2022).

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
