# Peer review of "Environment-Adaptive Object Detection Framework for Autonomous Mobile Robots"

_sensors, 2022, doi:10.3390/s22197647_

Round 1

Reviewer 1 Report

This paper proposed a systematic solution for object detection to solve the environment class imbalance problem, namely M-ODF. Overall, the work is interesting, well structured, well written, and important. The idea is in time and meaningful for robots that move through various environments to carry out missions. Additionally, as far as I know, this is the first paper on the environment class imbalance problem. However, there are some minor revisions needed.

1) In the caption of Figure 11, the explanations for (a) and (b) are reversed. ((a) detection accuracy for Top1 and Top5 and (b) CPU usage). 

2) It would be good to change the description of the vertical axis of the figure to 'occurrence frequency of class' to clarify.

3) The font of the text in the pictures is too small. In particular, it isn't easy to recognize what is being described in Figure 1 (a) and Figure 10.

4) Experiment results should be summarized in the abstract simply.

Reviewer 2 Report

The paper introduces a framework for mobile robots to detect objects with an aim to improve object recognition accuracy by using a model trained for the environment in which a robot is located at a given time.  The framework includes multiple models (each for a different environment) and a scheme to detect changes in environment, triggering a change in model.

The overall M-ODF framework has three main components:  multiple models for object detection, lightweight scene classification, and transition-probability-based model caching.  Each of these is OK individually, but the combination makes a nice contribution.  The models specific to environmental contexts aim to improve accuracy.  The scene classification and model caching aim to make computationally efficient the tasks of determining when a robot moves into a new environment and of switching models when that happens.  Experimental results document each of these benefits.
